# Targeting Abdominal Obesity and Its Complications with Dietary Phytoestrogens

**DOI:** 10.3390/nu12020582

**Published:** 2020-02-23

**Authors:** Alina Kuryłowicz, Marta Cąkała-Jakimowicz, Monika Puzianowska-Kuźnicka

**Affiliations:** 1Department of Human Epigenetics, Mossakowski Medical Research Centre, Polish Academy of Sciences, 5 Pawinskiego Street, 02-106 Warsaw, Poland; mcjakimowicz@imdik.pan.pl (M.C.-J.); mpuzianowska@imdik.pan.pl (M.P.-K.); 2Department of Geriatrics and Gerontology, Medical Centre of Postgraduate Education, 61/63 Kleczewska Street, 01-826, Warsaw, Poland

**Keywords:** phytoestrogens, diet, obesity, adipose tissue, metabolic syndrome

## Abstract

In the assessment of the health risk of an obese individual, both the amount of adipose tissue and its distribution and metabolic activity are essential. In adults, the distribution of adipose tissue differs in a gender-dependent manner and is regulated by sex steroids, especially estrogens. Estrogens affect adipocyte differentiation but are also involved in the regulation of the lipid metabolism, insulin resistance, and inflammatory activity of the adipose tissue. Their deficiency results in unfavorable changes in body composition and increases the risk of metabolic complications, which can be partially reversed by hormone replacement therapy. Therefore, the idea of the supplementation of estrogen-like compounds to counteract obesity and related complications is compelling. Phytoestrogens are natural plant-derived dietary compounds that resemble human estrogens in their chemical structure and biological activity. Supplementation with phytoestrogens may confer a range of beneficial effects. However, results of studies on the influence of phytoestrogens on body composition and prevalence of obesity are inconsistent. In this review, we present data from in vitro, animal, and human studies regarding the role of phytoestrogens in adipose tissue development and function in the context of their potential application in the prevention of visceral obesity and related complications.

## 1. Introduction

Obesity, which affects over 1.9 billion individuals worldwide and is responsible for 2.8 million deaths annually, has reached pandemic proportions [1]. Obesity-related diseases, e.g., hypertension, type 2 diabetes mellitus (T2DM), and dyslipidemia (the prevalence of which increases with increasing adipose tissue content), additionally impair one’s quality of life and shorten one’s lifespan. Despite constant progress in understanding its pathogenesis, its therapeutic potential to prevent and combat obesity is limited. Therefore, there still is a need for new therapeutic options for the prevention and treatment of obesity.

In the assessment of the health risk of an obese individual, both the amount of adipose tissue and its distribution and metabolic activity are essential. In humans, three distinct types of adipocytes have been identified: white (WAT), brown (BAT), and beige, which differ in their metabolic profiles [2]. In adults, the most abundant is WAT, which is responsible for the storage of the excess energy and the secretion of various mediators that contribute to the development of obesity-related complications. The metabolic and secretory activity of WAT may differ depending on its location—visceral (VAT) or subcutaneous (SAT) [3]. The distribution of WAT was found to be a predictor of health risk, and it differs in a gender-dependent manner. Overall, women have a higher adiposity than men. Men are prone to visceral (android) obesity that is associated with increased cardiometabolic risk. In women, most adipose tissue localizes in the subcutaneous depot (gynoid obesity) and, based on clinical observations, seems to be less harmful in the context of the development of obesity-related complications [4,5]. This hypothesis has been confirmed in animal models, where the transplantation of the SAT from donor mice into the visceral region of recipient mice has been found to lead to a decrease in total fat content and improves glucose homeostasis, thus suggesting that adipocytes from SAT excrete factors that, by acting systemically, improve metabolism [6]. The sexual dimorphism of adipose tissue distribution appears during puberty, which indicates the role of sex hormones in this phenomenon [7]. In adults, the gender-related difference in the distribution of adipose tissue persists and is regulated by sex steroids, especially estrogens [7]. In addition, sex steroids modulate adipocyte differentiation and are also involved in the regulation of lipid metabolism, insulin resistance, and inflammatory activity of the adipose tissue [8]. On the other hand, their deficiency contributes to the occurrence of metabolic diseases. For instance, the decrease in estrogen levels that has been observed in menopausal women has been found to result in an increase of the VAT volume and a risk of metabolic complications that can be partially reversed by hormone replacement therapy [9]. Therefore, the idea that the supplementation of estrogen-like compounds can counteract obesity and related complications is plausible.

Phytoestrogens are natural plant-derived dietary compounds that resemble human estrogens in their chemical structure and biological activity. For this reason, they were initially classified as endocrine-disrupting chemicals (EDCs). A more modern approach classifies phytoestrogens according to their primary mechanism of action as selective estrogen modulators (SERMs) or selective tissue estrogen action regulators (STEAR) [10,11,12]. Supplementation with phytoestrogens may confer a range of beneficial effects [13,14]. These include, but are not limited to, the alleviation of symptoms that are associated with estrogen deficiency in women, the improvement of insulin sensitivity, glucose and cholesterol metabolism, all resulting in a reduced risk of atherosclerosis [15]. However, the results of studies on the influence of phytoestrogens on body composition and prevalence of obesity are inconsistent, suggesting that their action on adipose tissue metabolism and distribution is complex and can be modified by other factors. In this review, we present data from the in vitro, animal, and human studies regarding the role of phytoestrogens in adipose tissue development and function in the context of their potential application in the prevention of obesity and related complications. 

## 2. Groups, Classes and Natural Sources of Phytoestrogens

The term “phytoestrogen” was developed in 1926 (from the Greek words phyto—“plant”—and estrogen—“sex hormone”) and refers to a functional, not chemical, classification of compounds. Phytoestrogens belong to polyphenols and, based on their chemical structure, are divided into two main groups: flavonoids (including isoflavones and coumestans) and non-flavonoids (including lignans and resorcinol derivates) [16]. 

Flavonoids are polyphenols that are synthesized exclusively by plants and are present, for example, in fruits, vegetables, tea, or ginger root. Isoflavones are the largest subgroup of flavonoids and possess phytoestrogen activity. They can be found in the Leguminosae family (especially in soy) as non-active hydrophilic glycosides (e.g., daidzin, genistin, and glyctin) and 4’-methylated lipophilic derivatives (e.g., formononetin and biochanin A) [17]. 

In the gastrointestinal tract, under the influence of the bacterial β-glucosidases, non-active isoflavones are hydrolyzed into bioactive aglycones, e.g., daidzin to daidzein, genistin to genistein, and glyctin into glycitein. While some other aglycones are resistant to degradation and are absorbed in the colon, daidzein, genistein, and glyctin undergo further transformation by intestinal flora. For example, daidzein can be metabolized to equol, a metabolite with a strong estrogenic and antioxidant activity, as well as to other, less active derivates. Individuals differ in their ability to transform isoflavones to particular metabolites, depending on their microbiome composition. Isoflavones and their metabolites have a short half-life (about nine hours for daidzein and seven hours for genistein) and are eliminated with urine. Urinary concentrations of equol are used for the screening of the soy protein dietary intake [18,19].

The second subgroup of flavonoids constitutes of coumestans, which exert the most pronounced estrogenic effect of all phytoestrogens. The best-known representative of coumestans is coumestrol, which can be found in red clover, spinach, brussels sprout, and legumes such as soybeans. Other examples of coumestans are repensol and trifoliol, which are present in split peas, pinto beans, lima beans, and clover sprouts [20]. 

Lignans are representatives of non-flavonoid phytoestrogens with weak estrogenic activity. They are components of the plant cell wall and can be found, for example, in oilseeds such as flaxseed, sesame, soy, and rapeseed, as well as whole-grain wheat, oats, rye cereals, and various vegetables. After consumption, plant lignans, secoisolariciresinol, and matairesinol are transformed by gut microflora into enterolignan metabolites, enterodiol, and enterolactone [21]. Urinary enterolignan metabolite concentrations may serve as a marker of lignan intake and the gut microbial environment.

Another representative of non-flavonoid phytoestrogens is resveratrol (3,5,4’-trihydroxy-trans-stilbene (RSV)). It is present in grapes, berries, peanuts, and selected teas. After ingestion, RSV is absorbed into the enterocytes and undergoes conjugation with sulfate and glucuronate. RSV sulfates and glucuronides are either transported to the bloodstream or, through the apical membrane, into the intestinal lumen, where they can be further metabolized in the colon by the gut microbiota. The microbial metabolites of RSV subsequently enter the circulation and act on different tissues and organs. The pleiotropic effects of RSV and its metabolites on human organisms include, but are not limited to, estrogenic, antioxidant, vasodilatory, anti-aging, antiviral, antidiabetic, and antiplatelet activities [22]. The classification of phytoestrogens is summarized in Figure 1.

## 3. Modes of Phytoestrogen Action

Phytoestrogens exert their action by modulating various cellular pathways. Their mechanisms of action can be divided into those that are related to the modulation of transcriptional activity of nuclear receptors and those that are not associated with the nuclear receptor signaling.

### 3.1. Modulation of Transcriptional Activity of Nuclear Receptors 

Due to their structural similarity to 17-β-estradiol (E2), the main female sex hormone, the natural and primary target for phytoestrogens are estrogen receptors (ERs) [23]. ERs exist in two main forms, α and β, which have multiple splice variants and exhibit tissue specificity in their expression and function. ERα seems to have a stronger physiological role in females than in males, while the role of ERβ does not seem to depend on sex. The classical, genomic action of ERs involves their conformational change in response to ligand binding, the release of chaperone proteins, the formation of homodimers, and the binding to estrogen response elements (ERE) in the target gene promoters or other response elements through association/tethering with other transcription factors that are bound to DNA. The recruitment of coactivators and interactions with other components of the basal transcription complex lead to the induction/repression of the transcription of target genes [24]. Phytoestrogens differ in their affinity towards ERs and estrogenic potencies. In general, most phytoestrogens have lower estrogenic activity than E2 and a higher binding preference for the ERβ than for the ERα [23].

Given the different roles of ERα and ERβ in gene regulation, their varying ratios in tissues may influence the cellular response towards different phytoestrogens. Additionally, the physiological levels of ER can be modulated by different pathological conditions. In the context of adipose tissue, the levels of ERα and ERβ seem to be significantly lower in obese individuals compared to normal-weight subjects, and this phenomenon concerns both visceral and subcutaneous depots (Kuryłowicz, A, unpublished data). Other factors that may contribute to the final effect of phytoestrogen (agonistic or antagonistic) on ERs in a given tissue include the availability of the ER coactivators and corepressors, as well as the level of endogenous estrogens. One of the most profoundly investigated phytoestrogens in this aspect is genistein. Its concentration that is needed for the activation of ERβ is much lower than that which is necessary for the activation of ERα. Therefore, at higher levels that can activate ERα, ERβ is also activated, and this phenomenon is responsible for counteracting some ERα-mediated effects [23]. This biphasic effect of phytoestrogen action has been shown in the case of the genistein-regulated proliferation of cancer lines, as well as in studies on the differentiation of osteocytes and adipocytes [25,26].

ER-mediated mechanisms cannot solely explain the effects of phytoestrogens’ actions. Other targets for phytoestrogens are peroxisome proliferator-activated receptors (PPARs) [27]. PPARs are nuclear receptors that are present in three isoforms: PPARα, PPARδ, and PPARγ, which encoded by different genes. PPARs act as ligand-regulated transcription factors that control gene expression by binding to specific response elements within promoters. PPARs bind as heterodimers with a retinoid X receptor and, upon binding to an agonist, interact with cofactors; this results in an increase of the transcription rate [28]. Several phytoestrogens, including genistein, daidzein, and RSV, can bind to PPARγ, modulate its transcriptional activity, and, in this way, interfere with downstream cellular pathways. This mechanism of phytoestrogens’ action is of particular interest in the context of obesity and related complications. Firstly, PPARγ plays a central role in the regulation of adipogenesis and adipose tissue browning. Secondly, the activation of PPARγ improves insulin sensitivity and blood glucose homeostasis, which is used for the treatment of obesity-associated diabetes in humans [29]. Finally, it has been found that obesity does not impair PPARγ expression in adipose tissue (Kuryłowicz, A, unpublished data). This finding gives hope that in the future, phytoestrogens that are able to activate PPARγ might be used to treat human obesity.

Apart from the modulation of PPARγ, isoflavones such as formononetin, biochanin A, genistein, and daidzein also have the potential to activate PPARα, a chief regulator of genes that are involved in fatty acid beta-oxidation and vascular inflammation, as well as PPARδ, which can be activated by long-chain fatty acids and thus acts as fatty acid sensor that regulates a variety of genes that are implicated in lipid metabolism [30].

Another nuclear receptor that is targeted by phytoestrogens is the aryl hydrocarbon receptor (AhR). AhR is a ligand-activated transcription factor that, after dimerization with aryl hydrocarbon receptor nuclear translocator (ARNT), binds to xenobiotic response elements (XREs) and modulates the expression of downstream genes. AhR plays a role in cell cycle control and apoptosis, in the regulation of the immune, cardiovascular, and reproductive systems, and in the proper function of the liver. Some AhR ligands are classified as endocrine disruptors because, upon binding to AhR, they may crosstalk with steroid hormone receptors and disturb normal hormone pathways. Both formononetin and biochanin A have been found to be potent AhR activators, as was shown in an in vitro study on yeast cells [31]. 

Genistein can also activate nuclear respiratory factors (Nrfs), transcription factors that are essential for the regulation of antioxidative enzyme expression. Nrfs dimerize with the Maf protein, and the formed complex binds to the antioxidative response element (ARE) in the regulatory regions of target genes. The activation of these pathways increases cellular defense against the toxicity of the reactive oxygen species (ROS), as was shown on animal models of atherosclerosis and cultured vascular cells [32].

### 3.2. Nuclear Receptor-Independent Mechanisms of Phytoestrogen Action

Several studies have proven that phytoestrogens also act in mechanisms that are independent of nuclear receptors. This refers to their ability to activate membrane-associated forms of ERα that interact with other signaling molecules, e.g., G proteins, growth factor receptors, and tyrosine kinases (Src). Binding to the cell membrane and extranuclear ERs activates protein kinases that phosphorylate other transcription factors and promote their nuclear translocation and transcriptional action [33]. The ability to activate G protein-coupled estrogen receptors was proven for genistein and equol; however, the biological effects of such interactions may differ depending on the tissue type [34].

Moreover, genistein, via the reduction of ROS levels and the induction of antioxidant enzymes, can activate the adenosine monophosphate-activated protein kinase (AMPK), as was found in the adipose tissues of mice on a high soy-containing diet [35]. RSV, by interfering with mitochondrial function and the subsequent increase of the AMP/ATP ratio, can also activate AMPK [22]. Importantly, AMPK plays a central role in the regulation of energy metabolism, especially in the balance between the anabolic and catabolic states. This ability of AMPK gives hope that its activators might be useful in the treatment of obesity and related complications. AMPK is also associated with the wingless-related integration site (Wnt)/β-catenin pathway, which, through the inhibition of the CCAAT/enhancer-binding protein alpha (C/EBPα) and PPARγ, inhibits adipocyte differentiation [36,37,38]. 

Isoflavones also act as protein tyrosine kinase inhibitors, which makes them potent regulators of different signaling pathways both in health and disease. An example of this kind of action is the effect of genistein on extracellular signal-regulated kinases 1 and 2 (ERK1/2) activity. ERK1/2 are members of the mitogen-activated protein kinases (MAPKs) that are involved in signaling cascades that regulate several cellular functions, such as cell proliferation and differentiation [39]. Some isoflavones inhibit DNA topoisomerases I and II, as well as ribosomal S6 kinase, while RSV can directly activate sirtuins, the nicotinamide adenine dinucleotide(NAD)-dependent deacetylases that, by modification of histones and transcription factors, control the expression of other genes [40]. However, it should be stressed that the described above mechanisms of phytoestrogen action are dose-dependent, and the concentrations of compounds that are used in most of the in vitro experiments were substantially higher than those found under physiological conditions in human plasma [41].

Some phytoestrogens may interfere with the activity of enzymes that are involved in estrogen metabolism. Coumestrol, due to its structure (two hydroxyl groups in the same position as 17β-estradiol), can inhibit the activity of 3β-hydroxysteroid dehydrogenase, an enzyme that catalyzes the synthesis of progesterone from pregnenolone, 17α-hydroxyprogesterone from 17α-hydroxypregnenolone, and androstenedione from dehydroepiandrosterone, thereby modulating the biological effects of endogenous steroids, as was shown in human lung tissue [42]. Moreover, both enterolactone and coumestrol decrease the activity of the cytochrome P450 aromatase (CYP19A1) that is responsible for the conversion of androgens to estrone or estradiol in human preadipose cell cultures [43]. Additionally, in in vitro experiments that were performed on mammalian intestinal cells, RSV was found to inhibit intestinal α-glucosidases, enzymes that are essential for the release of glucose from more complex carbohydrates, resulting in the decrease of postprandial glucose peaks and post-load insulin levels, which contributes to a beneficial influence of RSV’s on glycemic control [44]. 

Important mechanisms of phytoestrogen action are those that are related to their ability to modify the epigenome. Epigenetic modifications such as DNA methylation and microRNA (miRNA) interference have been emphasized as important mechanisms, which are activated via environmental stimuli, that regulate gene expression in adipose tissue. Methylation is a widespread feature of the genome. It is obtained through the addition of a methyl group to cytosine that is positioned next to guanine (CpG), and regions with a high occurrence of CpG dinucleotides, called CpG islands, are commonly present in the regulatory regions of genes. Methylation in the promoter region results in a change in gene expression that can be achieved by various mechanisms, e.g., by limiting the access of transcription factors to DNA or by the recruitment of co-repressors that alter chromatin structure [45]. Studies regarding the role of phytoestrogens in the modification of methylation patterns have not been univocal, and results have depended on the experimental model. In in vitro experiments, genistein, coumestrol, daidzein, and equol were found to decrease the methylation of several tumor suppressor genes in cancer cell lines [46], which was mediated by the inhibition of DNA methyltransferase (DNMT) activity [47]. However, in clinical trials, the administration of genistein, daidzein, and glycitein caused the hypermethylation of various genes that encode tumor suppressors [48]. 

In the context of obesity, so far, only genistein and RSV have been directly related to epigenetic changes. In animal models, maternal genistein supplementation has been found to protect against obesity in agouti mouse offspring due to the increase in DNA methylation upstream of the transcription start site of the *Agouti* gene [49]. Increased DNA methylation levels were also observed in the liver and muscle tissues from monkeys that were fed with genistein, and this was associated with increased insulin sensitivity [50]. In turn, RSV can influence DNA methylation and histone deacetylation via the activation of sirtuins, as has been shown, e.g., in the retinal pigment epithelial cells [51]. 

Polyphenols indirectly exert their effects through miRNA. miRNAs constitute a class of noncoding, endogenous, small RNAs that regulate gene expression by translational repression. By binding to the complementary sites in the target mRNA, miRNAs cause the translation repression, cleavage, deadenylation, and degradation of target mRNA. In recent years, significant progress has been made in understanding the role of miRNA in the regulation of the expression of various genes, including those that are related to adipocyte differentiation and function. miRNAs that have been implicated in adipogenesis and adipocyte metabolism were found to be differentially expressed in adipose tissue from obese subjects before and after weight loss compared to normal-weight controls, as well as in different adipose tissue depots [52]. The role of phytoestrogens in the regulation of miRNA in obesity and related complications has been less studied. However, the pretreatment of human umbilical vein endothelial cells with genistein was found to lead to the decreased expression of miR-34a and miR-155, which subsequently resulted in the downregulation of several genes that encode pro-inflammatory mediators (E-selectin, P-selectin, monocyte chemotactic protein-1 (MCP-1), interleukin-8 (IL-8), vascular adhesion molecule-1 (VCAM-1), and intercellular adhesion molecule-1 (ICAM1)) [53,54]. The molecular mechanisms that underly these phenomena included the upregulation of two suppressors of inflammation: sirtuin 1 and the suppressor of cytokine signaling-1 (SOCS-1) by miR-34a and miR-155, respectively [53,54].

By triggering the above-described mechanisms, phytoestrogens can modulate adipose tissue development, distribution, and metabolism. 

## 4. Role of Phytoestrogens in Adipose Tissue Development and Distribution 

### 4.1. Adipogenesis

The pro- or anti-adipogenic effects of phytoestrogens are related to their ability to activate or inhibit PPARγ signaling in adipocytes. The pro-adipogenic activities of genistein and daidzein result from their ability to activate PPARγ. On the other hand, RSV acts as a PPARγ antagonist. RSV exerts its antagonistic activity towards PPARγ through the direct interaction with the receptor [55] but also indirectly by inducing the PPARγ ubiquitination and its subsequent proteolysis [56]. It is also possible that the binding with RSV may modulate the interaction of PPARγ with other nuclear receptors. By triggering this mechanism, RSV inhibits the differentiation of cultured adipocytes and reduces the basal levels of adipogenesis, and it also antagonizes the effects of the pro-adipogenic agent rosiglitazone [57]. In animal models of obesity, RSV, by increasing basic metabolic rate and insulin sensitivity, has been found to successfully counteract the effects of a high-fat diet and to prevent ovariectomy-associated weight gain [58,59].

Genistein and daidzein, at concentrations found in human plasma, were found to be able to enhance the adipogenesis of murine proosteoblastic KS483 cells and mouse bone marrow cells and increase the expression of adipogenic genes in vitro [60]. This finding is consistent with the results of animal studies where genistein was found to promote adipogenesis and increased adipose tissue weight in rodents on a balanced diet [61,62]. However, in other experiments, genistein (alone or in combination with daidzein and glyctin) has been found to act as an inhibitor of the adipogenic differentiation of mouse 3T3-L1 cells (that have a fibroblast-like morphology and are used in research on adipose tissue), to decrease the adipogenic differentiation and maturation of bone marrow stromal cells, and to stimulate their differentiation into osteoblasts [63,64,65]. Moreover, the administration of genistein or its metabolite, orobol, to C57/BL6 mice (the inbred mouse model used in different research applications) was found to decrease fat-pad weight by 50% and to attenuate high-fat diet (HFD)-induced weight gain and lipid accumulation, respectively [66,67]. 

These discrepancies can be explained by the multidirectional influence of genistein on adipogenesis. Apart from the direct effect on PPARγ, genistein downregulates the ERK1/2 signaling pathway, as shown in bone marrow-derived mesenchymal stem cells. The ERK1/2 phosphorylation of PPARγ during the early stage of adipocyte differentiation promotes adipogenesis, but the activation of ERK1/2 during the late stage of adipogenesis might inhibit adipocyte differentiation [39]. Moreover, the PPARγ-dependent effects of phytoestrogens on adipogenesis can be modified by their ability to activate ERs [68]. The activation of ERs was found to repress adipogenic differentiation and maturation in mouse bone marrow stromal cells and to inhibit adipocyte differentiation, lipid accumulation, and the expression of adipocyte-specific genes in primary human adipocytes [63]. On the molecular level, the downregulation of adipogenesis-related genes by estradiol was found to be mediated through the inhibition of the PPARγ coactivator recruitment [69]. In animal studies, mice of both genders with a homozygous null mutation of ERα have been found to develop obesity due to the reduced energy expenditure in the absence of hyperphagia [7]. The role of ERβ in adipose tissue distribution has been less studied. The selective activation of ERβ in adipocytes in in vitro settings has been found to induce the expression of genes that are involved in WAT browning, that in the in vivo study, this activation was found to reduce body weight and fat mass in animals on a high-fat diet [70]. The impact of phytoestrogens on ER and PPARγ activity may depend on local estrogen availability, and this may explain the gender-specific effects of phytoestrogens on adipose tissue development and distribution. This phenomenon concerns, for example, daidzein that concurrently activates ERs and PPARγ, and the balance between different actions of ERs and PPARs determines the daidzein-induced differentiation of mouse bone marrow cells and mouse osteoprogenitor KS483 cells into adipocytes [60]. Therefore, the cross-talk between ERs and PPARs is crucial for the effects of daidzein on adipogenesis.

Experimental data regarding the influence of other phytoestrogens on adipogenesis have been more consistent. Biochanin A that suppressed in a dose-dependent manner via the inhibition of PPARγ, was found to differentiate primary rat adipose tissue-derived stem cells to mature adipocytes, thereby promoting bone formation [71]. In turn, formononetin, by the ROS-induced AMPK activation, was found to suppress the adipogenic differentiation of 3T3-L1 cells [38]. Several synthetic formononetin esters have been evaluated in the context of their influence on adipogenesis. Among them, the ZWA3-1 compound was found to exhibit the highest activity in inhibiting preadipocyte differentiation, while the ZWD1-6 compound was found to be the most effective in inhibiting preadipocyte proliferation [72]. Gomisin N, a lignan that is derived from *Schisandra chinensis*, was found to inhibit 3T3-L1 preadipocytes differentiation in vitro in a dose-dependent manner at an early adipogenic stage. This effect was obtained through the activation of AMPK but also through the inhibition of PPARγ [73].

Finally, the anti-adipogenic effects of RSV are mediated mainly by sirtuins that, through the epigenetic modification of histones and the regulation of transcription factors, control the critical steps of preadipocytes differentiation and proliferation, as was shown e.g., in bovine intramuscular adipocytes [74].

In summary, the influence of phytoestrogens on adipogenesis and adipose tissue distribution is complex and compound-specific. Moreover, the final effect of the investigated compound may depend on the balance between the triggered mechanisms.

### 4.2. Adipose Tissue Browning

Since BAT is characterized by a higher metabolic activity than WAT, the activation of genes that are specific to brown adipocytes in preadipocytes is one of the concepts of obesity treatment. White adipocyte browning is an adaptive and reversible process that can be triggered by various stimuli.

Due to its ability to activate sirtuins, RSV is probably the best-studied phytoestrogen in the context of adipose tissue browning [40]. When activated by RSV, sirtuin 1 upregulates the expression of genes that are typical for BAT and downregulates genes that are specific for WAT in 3T3-L1 preadipocytes [75]. The same effect was found to be induced by genistein, which also acts as a sirtuin 1 activator [76]. 

Both diets rich in isoflavones and the administration of daidzein were found to result in the increased expression of uncoupling protein 1 (ucp1), a thermogenic marker, in the BAT of experimental animals [77,78]. Additionally, formononetin, via direct interaction with PPARγ, was found to increase the expression of ucp1 in primary cultures of mouse adipocytes, while its administration to C57BL/6J mice was found to prevent the development of diet-induced obesity [79]. Similarly, a two-week treatment with coumestrol was found to reduce body weight and improved glucose tolerance in mice on an HFD due to the increased thermogenesis in BAT. On the molecular level, coumestrol was found to upregulate AMPK and sirtuin 1, and its beneficial effects were found to be inhibited by the ERα knock-out [80]. The only concern regarding these studies is that the dosage of phytoestrogens that was used in the experiments was much higher than the physiological range that was obtained from dietary intake. In this regard, future studies on the dose–response relationship are required for the development of phytoestrogen-based strategies that target adipose tissue browning.

## 5. Role of Phytoestrogens in Adipose Tissue Metabolism

Estrogen not only influences adipose tissue amount but also its metabolism and, therefore, modulates the risk of obesity-related complications. In clinical studies, menopause has been found to be associated with a constant decline in insulin sensitivity that is parallel to the increase in serum inflammatory markers and an unfavorable lipid profile [81]. In animals, ovariectomy has been found to lead to insulin resistance and increased susceptibility to the deleterious effects of a high-fat diet. This could be prevented by estrogen supplementation, thus resulting in its physiological concentrations [82]. In in vitro and in vivo experiments, estradiol was found to control the activity of lipoprotein lipase (LPL), an indicator of lipid storage, in a dose-dependent manner. In clinical trials, the transdermal administration of estradiol was found to decrease the expression of genes that encode critical lipogenic enzymes (stearoyl-CoA desaturase, fatty acid synthase, acetyl-coenzyme A carboxylase alpha, and fatty acid desaturase 1) in human SAT in a way that correlated with a decrease in plasma triglyceride (TG) levels [83].

Given the multitude of pathways that are modulated by phytoestrogens, it is plausible that their activities that resemble the action of estrogen in adipose tissue are not limited to the control of adipogenesis but also include the regulation of adipocyte metabolism and secretory activity. 

### 5.1. Lipogenesis and Lipolysis

The influence of isoflavones on gene expression in WAT was assessed in a clinical trial with two isoflavone supplements, one based on daidzein, the other one on genistein, that were administered to postmenopausal women for eight weeks. Daidzein-based supplementation, via the activation of PPARγ, was found to downregulate the expression of genes that are involved in lipid synthesis in the same way as caloric restriction in mice [84]. The same effect was observed during daidzein supplementation to C57BL/6J mice with diet-induced obesity [85]. What is essential is that this action of daidzein did not depend on the individual’s ability to produce equol, its active metabolite, as was shown in adipose tissue samples that were obtained from women on isoflavone supplementation [84]. Additionally, genistein and coumestrol were found to decrease basal and insulin-induced lipid synthesis in adipocytes that were isolated from the ovariectomized rats and in 3T3-L1 preadipocytes, respectively [86,87,88]. 

Apart from the inhibition of lipogenesis, several phytoestrogens have been found to induce lipolysis, triggering different molecular mechanisms. Genistein and daidzein were found to enhance lipolysis by suppressing cyclic adenosine monophosphate phosphodiesterase (cAMP-specific PDE) in rat adipocytes that lead to decreased triglyceride (TG) accumulation and prevented their hypertrophy [88,89]. In turn, coumestrol was found to reduce lipid accumulation in 3T3-L1 preadipocytes via the upregulation of genes that are related to fatty acids oxidation, such as hormone-sensitive lipase, carnitine palmitoyltransferase 1, and uncoupling protein 2. Subsequently, coumestrol’s administration to animals on an HFD has been found to increase lipolysis in white adipose tissue depots and, in this way, prevented obesity [80,87,88]. Additionally, formononetin was found to promote lipolysis and increase glycerol release in adipocytes [72]. Finally, RSV was found to increase lipolysis through the activation of sirtuins and the subsequent (i) deacetylation of PPARγ corepressors in a way that stimulated the transcription of the gene that encodes adipose triglyceride lipase; (ii) the downregulation of sterol regulatory element-binding proteins 1 and 2 (SREBP-1 and SREBP-2) transcription factors that are crucial for sterol biosynthesis; and (iii) the deacetylation of liver X receptors (LXR) that act as cholesterol sensors and regulate lipid homeostasis [90]. 

### 5.2. Adipose Tissue Inflammation and Secretory Activity 

Recent research has widened our understanding of the role of adipose tissue from that of pure energy storage to that of a metabolically and hormonally active organ that, in response to environmental stimuli, is able not only to activate lipolysis/lipogenesis but also to secrete several factors to communicate with and regulate the function of other organs. These findings have allowed us to understand the link between excess adiposity and the development of obesity-related complications and renewed interest in adipose tissue as a possible target for obesity-oriented therapies [91]. 

Obese individuals develop a low-grade chronic inflammatory state, also called metaflammation. Elevated levels of inflammatory markers correlate with an increased risk of obesity-related complications, including cardiovascular diseases, T2DM, and dyslipidemia, and they stimulate the infiltration of endothelium by lymphocytes, as well as vascular smooth muscle cells migration, promoting intimal thickening and atherosclerosis. Inflammation can, therefore, be a common pathomechanism that connects obesity with other components of the metabolic syndrome. In vitro experiments have suggested that the initiation of the inflammatory process in response to an excess of nutrients takes place in the adipose tissue itself. According to this theory, the accumulation of lipids leads to the increased expression of genes that encode cytokines, chemokines, and adhesion molecules in adipocytes, and this attracts infiltrating immune cells that further contribute to the synthesis of pro-inflammatory mediators [92].

The anti-inflammatory effects of isoflavones were demonstrated by in vitro experiments by using lipopolysaccharide-activated macrophages. Several studies have proven that phytoestrogens might also modulate the secretory profile of adipocytes [36]. 

Genistein, via the inhibition of nuclear factor κB (NF-κB, a protein complex that controls transcription of key genes that are involved in, among others, inflammatory response and apoptosis), downregulates synthesis of pro-inflammatory interleukins (e.g., IL-6 and IL-8) in mouse 3T3-L1 cells. At the same time, through the inhibition of JAK-2 (Janus kinase 2), a protein tyrosine kinase, genistein in human fibroblasts decreases the expression of leptin—a hormone and a pro-inflammatory cytokine secreted by adipocytes [65]. Genistein-based supplementation has also been found to lead to the activation of the upstream regulator interferon b1 (IFNB1). The activation of this cytokine inhibits the NOD-like receptor P1b and P3 (NLRP1b and NLRP3) inflammasomes (cytosolic multiprotein oligomers of the innate immune system that are responsible for the activation of inflammatory responses) via the upregulation of the signal transducer and the activation of transcription factor 1 (STAT1) [93]. However, these effects were only observed in equol producers. Additionally, biochanin A, via the inhibition of PPARγ and the blocking of MAPK phosphorylation, downregulates leptin, TNFα, and IL-6 expression in primary rat adipose-derived stem cells [71].

Daidzein regulates pro-inflammatory gene expression by activating PPARα and PPARγ, as well as by inhibiting the c-Jun N-terminal kinase (JNK) pathway in 3T3-L1 adipocytes and macrophages co-cultures. These effects include the decreased expression of MCP-1 and IL-6 and the increased expression of adiponectin [94]. The decreased expression of MCP-1 and TNF-α and the increased expression of adiponectin were also observed in adipose tissue of C57BL/6J daidzein-fed mice with diet-induced obesity, and this was accompanied by inhibited macrophage infiltration. In a clinical trial, daidzein-based supplementation significantly downregulated the expression of the *MAPK1* and *KRAS* genes, which are involved in the activation of the inflammatory response in SAT of postmenopausal women, with the simultaneous upregulation of genes that encode proteins with anti-inflammatory properties, such as IL-10 receptor antagonist (IL10RA) and NF-κB inhibitor α (NFKBIA) [84]. 

Through the upregulation of sirtuin 1, RSV represses NF-κB pro-inflammatory responses in adipocytes and macrophages that infiltrate adipose tissue, leading to an improvement in insulin signaling and insulin resistance [95]. Moreover, RSV favorably modulates the profile of adipokines that are secreted by adipose tissue; RSV increases the expression of anti-inflammatory adiponectin and omentin and decreases the expression of resistin and visfatin, which have pro-inflammatory properties [40].

The mechanisms of phytoestrogen action on adipose tissue development, metabolism, and secretory activity are summarized in Table 1.

## 6. Role of Phytoestrogens in the Management of Obesity and Related Complications

### 6.1. Influence of Phytoestrogens on Body Weight and Composition

The decline in estrogen levels during menopause is associated with an increase in the prevalence of obesity in women and changes in adipose tissue distribution from a gynoid to an android type. On the contrary, women who use hormone replacement therapy do not present the characteristic abdominal fat-related weight gain pattern that is usually associated with metabolic complications of obesity [97]. Given the role of phytoestrogens in the regulation of adipogenesis and adipose tissue metabolism and their molecular mechanisms of action, the idea of their administration to regulate body weight is, therefore, plausible. Moreover, since phytoestrogens can influence adipose tissue distribution, their supplementation might preferably decrease the amount of VAT and, in this way, reduce the risk of metabolic complications of obesity. 

Nevertheless, epidemiological studies that have linked dietary intake of phytoestrogens with adiposity have not been univocal. According to the World Health Organization, in a survey of 167 countries, soy consumption correlated positively with obesity, irrespective of caloric intake [98]. However, data on soy dietary intake in this report were generalized and approximate, based on the Food and Agriculture Organization resources. Moreover, studies based on the food frequency questionnaire (FFQ) are also are at risk of inaccuracy because they do not correspond to serum phytoestrogen levels [99]. 

On the contrary, more accurate studies have indicated the potentially protective effect of phytoestrogens against obesity. In 6806 participants of the National Health and Nutrition Examination Survey (NHANES) 2003–2008 study, high urinary enterolignans (enterodiol and enterolactone) concentrations were associated with a lower likelihood of being obese (by 42% and 64%, respectively). Moreover, high urinary enterodiol levels were associated with a 48% lower likelihood of having high-risk waist circumference (a marker of visceral obesity) in adult participants of this study [100]. In turn, in 1273 male adult participants of the NHANES 2001–2010 study, urinary enterolactone levels were inversely associated with obesity and the prevalence of the metabolic syndrome components [101], while, in the NHANES 2003–2010 study, also with abdominal obesity [102]. Similar conclusions came from the Framingham Offspring Study, where postmenopausal women with a high dietary intake of lignans had a significantly lower waist–hip ratio (WHR) compared to the participants who declared lower lignans consumption [103]. While the NHANES 2001–2010 study detected no associations with urinary daidzein, equol, or genistein and anthropometric parameters [101], in another cross-sectional study, postmenopausal women with a high (≥1.0 mg/day) genistein intake had a significantly lower body mass index (BMI) and waist circumference than those with no daily genistein consumption [104]. However, the major limitation of the last study was a low number of participants (*n* = 208).

Therefore, the results of these epidemiological studies have suggested that the increased consumption of phytoestrogens can decrease the amount of VAT and, in this way, reduce the risk of metabolic complications of obesity. Additionally, some above-mentioned animal studies have indicated the potential therapeutic use of particular compounds in visceral obesity treatment. For example, the 12-week administration of formononetin was found to inhibit the development of HFD-induced obesity in mice with a specific reduction of visceral fat accumulation. Moreover, changes in adipose tissue mass and distribution were found ot be accompanied by the improvement of lipid serum profile, namely an increase in high-density lipoprotein (HDL) cholesterol and a decrease in TG levels, as well as a reduction of liver steatosis [38]. These findings inspired researchers to test the anti-obesity potential of phytoestrogens in clinical intervention trials.

However, the available results of individual randomized controlled trials (RCTs) that assessed the effects of phytoestrogen supplementation on body weight are contradictory, most possibly due to the heterogeneity of these studies. The studies differed in the participants’ number and characteristics (healthy vs. with metabolic disorders), as well as the type of the intervention (compound, dosage, time of intervention). For instance, isoflavone mixture supplements have been found to significantly decrease body weight in healthy postmenopausal women. In contrast, the administration of isoflavones rich in daidzein was found to be associated with an increased body weight in women suffering from metabolic disorders [105,106]. Moreover, it should be remembered that the effectiveness of phytoestrogen supplementation depends on the intestinal microflora composition that is responsible for their transformation into active metabolites [107]. In equol non-producers, overweight and obesity are diagnosed more frequently, and supplementation with isoflavones is less effective in the improvement of serum glucose and low-density lipoprotein cholesterol (LDL-C) levels [108]. 

In rodents, RSV has also been found to be able to decrease adiposity by inhibiting fat accumulation and stimulating the lipolytic pathways. Therefore, RSV effectiveness has also been tested in several clinical trials. However, none of them have resulted in significant weight reductions [109,110,111,112,113,114].

Apart from single clinical trials, during the last seven years, two meta-analyses assessed the influence of dietary phytoestrogens on body composition in women. A meta-analysis of nine randomized controlled trials, all conducted in postmenopausal non-Asian women, 272 on soy isoflavone supplementation, and 256 controls, suggested that phytoestrogens may help to reduce body weight. The dose varied from 40 to 160 mg of a compound daily, while the intervention time varied from 8 to 52 weeks. Interestingly, the beneficial effect on body weight was more pronounced in the shorter (<6 months) duration group in those who consumed lower dose (<100 mg) of isoflavones daily and those who were non-obese (BMI < 30 kg/m^2^) at baseline [115]. Based on these results, the authors concluded that soy isoflavone supplementation might be more effective in weight reduction in non-obese women with a lower dose of isoflavone and in a shorter time of supplementation.

A more recent meta-analysis included 23 RCTs and 1880 postmenopausal women of different ethnicities, 1130 in the intervention group, and 750 controls. The type of administered compound varied from the mix of isoflavones and lignans to single genistein or daidzein; the intervention time ranged from 8 to 48 weeks. The overall analysis revealed an association between phytoestrogen supplementation and a decrease in the WHR that corresponded to a lower cumulation of visceral fat. In contrast, no significant changes in body weight, BMI, waist and hip circumference, total fat mass, or percentage of body fat were observed [106]. However, in a subgroup that only included healthy individuals who were on isoflavone mix, a modest decrease in body weight after the intervention was found. On the contrary, in women with obesity-related disorders (T2DM, hypertension, and hyperlipidemia), phytoestrogen supplementation was associated with an increase in body weight. Moreover, a meta-analysis indicated that daidzein, in contrast to soy products or isoflavone mix, could lead to unfavorable changes in body composition. Therefore, the authors concluded that the influence of phytoestrogens on body weight is compound-specific and depends on the underlying metabolic status.

The discrepancies between both meta-analyses could have resulted from variations in study protocols that included (i) different types of phytoestrogen supplements (soy-derived isoflavones vs. isoflavones and lignans); (ii) the baseline characteristics of the study participants (ethnicity, health status); (iii) the number of individuals under study. However, both of them suggested that, under certain circumstances, phytoestrogen supplementation may contribute to weight and visceral adipose tissue reduction. Therefore, since phytoestrogens favorably affect not only the amount of adipose tissue but also its metabolism and secretory profile, they might be efficient in the treatment of visceral obesity-related complications [17], at least in selected sub-groups of patients. 

### 6.2. Influence of Phytoestrogens on Hyperlipidemia and Liver Steatosis

Epidemiological data have linked higher phytoestrogen consumption/metabolism with beneficial changes in blood lipid profiles. In male participants of the NHANES 2001–2010 study, aged 20–60 years, higher enterolactone urinary levels were inversely associated with TG levels and metabolic syndromes but positively associated with HDL-cholesterol levels [101]. The analogous correlations were observed in the NHANES 1999–2004 and NHANES 2003–2010 studies for both genders [102,116]. In the Framingham Offspring Study, the high dietary intake of isoflavones and lignans was associated with significantly lower plasma TG levels [103], while in a cross-sectional study that assessed the influence of isoflavone intake on cardiovascular disease risk factors, a diet rich in daidzein was positively associated with HDL cholesterol in postmenopausal women [104]. Moreover, animal studies on dietary interventions with phytoestrogens in order to improve serum lipid profile or liver steatosis have provided promising results. Supplementation with daidzein, genistein, and glyctin was found to significantly reduce plasma TG in Sprague Dawley rats [117,118]. Genistein alone was found to decrease the incorporation of glucose into lipids and to increase the output of fatty acids from the liver of the ovariectomized rats that resulted in the diminution of the liver TG contents [86], while daidzein was found to lower total plasma cholesterol (TC) levels in Syrian hamsters [119]. However, in lean and obese SHR/N-cp rats (that serve as a genetically obese model for studies on non-insulin-dependent diabetes mellitus) on a diet that contained a 0.1% soybean isoflavone mixture, the TG level was found to be decreased only in lean animals [120]. The results of clinical trials have proven the beneficial effects of RSV on lipid profiles. RSV’s reformulated version, resVida, was found to significantly decrease TG levels and liver steatosis in obese men [109], while SRT2104, another RSV analog, was found to cause a decrease in serum TC and TG levels, as well as a significant reduction of the inflammatory parameters [121]. In meta-analyses of clinical trials, dietary soy intake has been found to exert a beneficial effect on TC, TG, LDL-cholesterol, and HDL-cholesterol concentrations [122].

Data regarding the influence of phytoestrogens on liver steatosis have been less consistent. While a high-isoflavone soy protein isolate diet was found to protect obese Zucker rats from liver steatosis, supplementation with daidzein alone did not improve their liver steatosis scores or body weight, energy intake, and serum leptin and adiponectin levels [123]. However, genistein via the phosphorylation of AMPK ameliorates fat accumulation in buffalo rat liver (BRL) cells, which makes it a potential candidate for the treatment of non-alcoholic fatty liver disease (NAFLD) [124]. Similarly, biochanin A was found to prevent obesity-induced hepatic steatosis and insulin resistance in HFD mice [125]. Some studies have reported the analogical effects of formononetin supplementation [126], while its synthetic analog, 2-heptyl-formononetin, was found to induce hepatic steatosis in C57BL/6J mice [127].

### 6.3. Influence of Phytoestrogens on Glucose Metabolism and Diabetes

The favorable influence of genistein, daidzein, and glyctin on lipid profile has been found to translate into an increased insulin sensitivity and lower plasma glucose and insulin levels in Syrian hamsters, a mouse model of nongenetic T2DM and both lean and obese rats [84,119,120,128]. Moreover, these isoflavones can stabilize β-cell function and postpone the onset of diabetes in non-obese diabetic (NOD) mice [129], while genistein alone could also do these things in streptozotocin (STZ)-induced diabetic mice [130]. The mechanisms of these actions are complex and include the activation of calmodulin kinase II and Ca^2+^ signaling and downregulation of the NF-κB, ERK-1/2, and JAK/STAT pathways [17]. 

In a cross-sectional study in postmenopausal women, genistein, daidzein, and total isoflavone intake were found to be inversely associated with fasting and post-challenge insulin levels [104]. In participants of the NHANES 2001–2010 study, urinary concentrations of total isoflavone metabolites were found to be inversely associated with the fasting glucose, insulin and homeostatic model assessment of insulin resistance (HOMA-IR) in pregnant women, while enterolactone urinary levels were found to be inversely associated with fasting glucose and insulin levels in males who were aged 20–60 years [101,122]. Dietary lignans consumption was found to be inversely associated with fasting insulin and C-peptide levels in men [131]. These findings suggest that isoflavones may have a direct effect on β-cell function and insulin secretion, which is supported by experimental studies. However, in the Singapore Chinese health study, including 564 diabetic patients and 564 healthy controls, no significant association between urine phytoestrogen metabolites and the risk of T2DM was found [132]. The discrepancy between the results of these studies may have resulted from the regional differences in their dietary intake. The consumption of soy products is generally lower in the Western diet, leading to the modest effect of isoflavones on metabolic markers. In the Western diet, lignans, not isoflavones, are the primary form of phytoestrogens.

The influence of phytoestrogens on glucose metabolism and diabetes risk has also been assessed in other clinical trials. Notably, the effects have been found to be sex-specific, which may have resulted from the estrogen-like activity of the compounds. Soy protein consumption (8 g/d) was found to be inversely associated with glycosuria in normal-weight postmenopausal women from the Shanghai Women’s Health study [133]. On the contrary, a similar soy protein intake was found to be associated with an increased risk of hyperglycemia in men [134]. Additionally, supplementation composition, as well as its duration, might have influenced the findings. In postmenopausal women with T2DM, one-year isoflavone intake (100 mg of aglycones) was found to improve insulin sensitivity and lipid profiles [135], while three-month supplementation with 132 mg isoflavones was not found to result in a decrease of plasma glycated hemoglobin A1c (HbA1c), blood glucose, or insulin levels [136]. 

In a meta-analysis that covered 10 randomized controlled trials and 794 perimenopausal and postmenopausal non-Asian women, soy isoflavone intake was not found to be associated with a significant glycemia reduction. However, it was found to improve insulin secretion and HOMA-IR [137]. The impact of phytoestrogen supplementation on glucose and insulin status was also assessed in the mentioned-above meta-analysis [106]. Phytoestrogen intake was associated with lower fasting glucose levels independently of weight, and the effect was more pronounced in a low-dose (< 100mg/d) but longer (≥ 6 months) treatment. Supplementation was also associated with lower baseline insulin, and the effect did not depend on the dose and time of treatment; however, it was more pronounced in the non-obese study participants.

Data regarding the influence of phytoestrogen on visceral obesity and its metabolic complications are summarized in Table 2. 

## 7. Final Remarks and Conclusions

Despite the constant progress in understanding the pathogenesis of obesity, therapeutic options are limited, and there is a need for new treatments. Due to their significant role in adipocyte development and metabolism, phytoestrogens might be an option for treating obesity and its metabolic complications.

However, there are some essential issues that should be addressed before phytoestrogen-based therapies implementation. Firstly, the results of different studies (epidemiological, in vitro, in vivo, and clinical trials) regarding the potential impact of phytoestrogens and its derivatives on obesity have not been univocal and have often been challenging to interpret. This fact reflects the complexity of phytoestrogens’ actions but has also resulted from the methodological differences between the studies. Though, in general, the epidemiological data point to a beneficial effect of phytoestrogen dietary intake on obesity and visceral adipose tissue depot, the main objections regarding these kinds of studies are related to their high heterogeneity. Dietary intake differs among populations and nationalities, as Asians generally consume higher amounts of isoflavones, while lignans constitute primary dietary phytoestrogens in Western populations. Apart from the diet, final phytoestrogen serum levels are determined by the metabolic activity of the microbiome [138]. Due to the high diversity in phytoestrogen consumption and inter-personal variations in their metabolism, it is difficult to define what concentrations of each compound should be considered “physiological” or “therapeutic,” as well as what levels reflect “deficiency.” This fact has implications for the construction of in vitro and in vivo experiments. Finally, the significant, interindividual differences in response to active phytoestrogen metabolites have been observed. These might have resulted from the local factors that include, for example, the in vivo estrogen environment and genetics (polymorphic variants of genes that encode target proteins, including ERs) [84]. Moreover, clinical trials have suggested that underlying diseases may modify the effectiveness of phytoestrogens in reducing body weight and may even lead to adverse effects of such treatment. It is suggested that to obtain the impact on body weight, supplementation with phytoestrogens should be accompanied by a hypocaloric diet and enhanced physical activity to maintain healthy body weight during the supplementation period [95].

In conclusion, despite promising results of in vitro and in vivo experiments, as well as some clinical studies regarding the application of phytoestrogens for the treatment of visceral obesity and its metabolic complications, there is a need for well-designed clinical trials that would help to assess what kind of intervention (type and dose of compound, duration) and in which individuals (ethnicity, morbidity, time since menopause onset, metabolic status, body composition, and diet composition at baseline) should be considered. Additionally, variations in phytoestrogen metabolism that are related to microbiome composition should be taken into account. 

## Figures and Tables

**Figure 1 nutrients-12-00582-f001:**
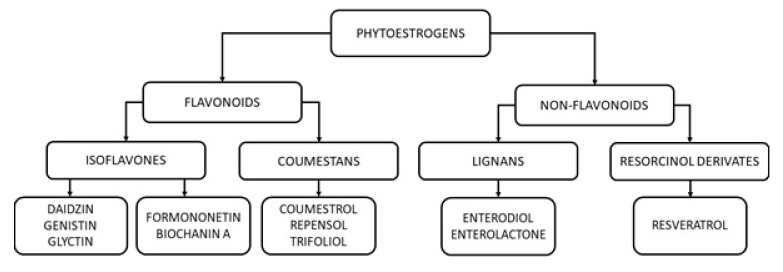
A simplified classification of phytoestrogens.

**Table 1 nutrients-12-00582-t001:** Influence of phytoestrogens on adipose tissue development, metabolism, and secretory activity.

			Influence on			Experimental Model	
Group of Phytoestrogens		Mechanism of Action	Adipogenesis	Lipolysis/Lipogenesis	Inflammation		References
Isoflavones	daidzein	↑PPARγ	↑ adipocytes differentiation ↑ adipocytes browning	↓lipids synthesis		KS483 cells 3LT3 cells rats on an HFD C57BL/6 mice human AT	[60] [76] [77] [83] [84]
		↑PPARγ ↑PPARα ↓JNK pathway ↓*MAPK1* ↓*KRAS*			↓MCP-1 ↓ IL-6 ↑adiponectin ↑IL10RA	3LT3 cells and RAW264 macrophages	[94]
		↑ERα	↓ adipocytes differentiation			3LT3 cells	[30]
		↓cAMP PDE		↑lipolysis		rat adipocytes	[89]
	genistein	↑AMPK	↓ adipocytes differentiation			CD1-mice	[37]
		↑PPARγ	↑ adipocytes differentiation	↓lipids synthesis		C57BL/6 mice C57BL/6 mice ovariectomized rats	[61] [85] [86]
		↓ERK1/2	↓ adipocytes differentiation			mice on LFD human adipocytes human PBMSC	[62] [63] [64]
		↑sirtuin 1	↑ adipocytes browning			3LT3 cells	[76]
		↓cAMP PDE		↑lipolysis		rat adipocytes	[96]
		↓NF-κB			↓IL-6 ↓IL-8 ↓leptin ↑ IFNβ	human fibroblasts and 3LT3 cells	[65]
		↓miR-155 ↓miR-34a			↓ E-selectin ↓ P-selectin ↓VACM-1 ↓ICAM-1 ↓MCP-1 ↓IL-8	HUVECs	[53] [54]
	biochanin A	↓PPARγ	↓ adipocytes differentiation		↓leptin, ↓TNFα ↓IL-6	yeast cells rat adipocytes	[31] [37]
	formononetin	↑AMPK	↓ adipocytes differentiation			C57BL/6 mice	[38]
		↑ PPARγ	↑ adipocytes browning			C57BL/6 mice	[79]
				↑lipolysis		3LT3 cells	[72]
	formononetin esters	↑AMPK	↓ adipocytes differentiation ↓ adipocytes proliferation			3LT3 cells	[72]
Coumestans	coumestrol	↑AMPK ↑sirtuin 1	↑ adipocytes browning	↑lipolysis		mice on an HFD 3LT3 cells	[80] [87]
				↓lipids synthesis		rat adipocytes	[88]
Lignans	gomisin N	↑AMPK ↓PPARγ	↓ adipocytes differentiation			3LT3 cells	[73]
Resorcinol derivates	resveratrol	↑AMPK ↓PPARγ	↓ adipocytes differentiation ↓ adipocytes proliferation			3LT3 cells 3LT3 cells mice on an HFD ovariectomized rats	[56] [57] [58] [59]
		↑sirtuins	↑ adipocytes browning	↑lipolysis	↑adiponectin ↑omentin ↓ resistin ↓ visfatin	mice on an HFD	[75]

AMPK—adenosine monophosphate-activated protein kinase; AT—adipose tissue; cAMP PDE—cAMP phosphodiesterase; ERα—estrogen receptor α; ERK1/2—extracellular signal-regulated kinases 1 and 2; HFD—high fat diet; HUVECs—human umbilical vein endothelial cells; ICAM1—intercellular adhesion molecule-1; IFNβ—interferon β; IL—interleukin; IL10RA—IL-10 receptor antagonist; JNK*—*c-Jun N-terminal kinase; *KRAS**—*Kirsten rat sarcoma viral oncogene homolog; LFD—low fat diet; *MAPK1—*gene encoding mitogen-activated protein kinase 1; MCP-1—monocyte chemotactic protein-1; NF-κB—nuclear factor κB; PBMSC—primary bone marrow stromal cells; PPAR—peroxisome proliferator-activated receptor; TNFα—tumor necrosis factor α; VCAM-1—vascular adhesion molecule-1; ↓ decrease; and ↑ increase.

**Table 2 nutrients-12-00582-t002:** Influence of dietary phytoestrogens on visceral obesity and its metabolic complications.

	Influence on									
Phytoestrogen	BMI	References	Visceral Obesity	References	Glucose Metabolism	References	Serum Lipids	References	Liver Steatosis	References
daidzein	↔	[101] [123]			↓glucose ↓insulin	[104]	↑HDL-C ↓TG ↓TC	[104] [123] [119]	↔	[123]
	↑	[106]					↑HDL-C ↓LDL-C ↓TG ↓TC	[118]		
genistein	↔	[101]	↓WC	[104]	↓DM progression ↓glucose ↓insulin	[104] [130]	↓TG	[38]	↓	[86] [125]
	↓	[104]								
formononetin	↓	[38]	↓VAT	[38]	↓IR	[125]	↓TG ↑HDL-C	[38]	↓	[38] [125,126]
									↑	[127]
soy isoflavones	↓	[115] [137]			↓↑↔ glucose ↓↔ insulin ↓IR ↓glycosuria	[104] [122] [133,134,135,136,137]	↑HDL-C ↓LDL-C ↓TG ↓TC	[124]		
soy isoflavones and lignans	↔	[106]	↔ WC ↓ WHR	[106]	↓glucose ↓insulin ↓IR	[119,120] [128] [85]	↓TG	[103]		
lignans					↓insulin ↓C-peptide	[131]				
enterolactone*	↓	[100,101]	↓ WC	[102]	↓glucose ↓insulin	[101]	↓TG ↑HDL-C	[101] [102]		
			↓ WHR	[103]						
enterodiol*	↓	[100]	↓ WC	[100]						
			↓ WHR	[103]						
resveratrol	↔	[109] [110,111,112,113,114]					↓TG	[109]	↓	[109]
							↓TC	[121]		

BMI—body mass index; DM—diabetes mellitus; HDL-C—high density lipoprotein cholesterol; IR—insulin resistance; LDL-C—low density lipoprotein cholesterol; VAT—visceral adipose tissue; TC—total cholesterol; TG—triglycerides; WC—waist circumference; WHR—waist–hip ratio; ↓ decrease; ↑ increase; ↔ no change, and * urinary. Clinical and epidemiological studies in humans: 100, 101, 102, 103, 104, 106, 109, 110, 112, 113, 114, 115, 121, 122, 131, 133, 134, 135, 136, 137. Studies on animal models of obesity: 38, 85, 86, 111, 117, 118, 119, 120, 123, 125, 126, 127, 128, 130.

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
