# Peer review of "Targeting Abdominal Obesity and Its Complications with Dietary Phytoestrogens"

_nutrients, 2020, doi:10.3390/nu12020582_

Round 1

Reviewer 1 Report

The article is interesting and contributes to the increase of knowledge in the field of phytoestrogens and abdominal obesity.

Author Response

Reviewer 1

  • The article is interesting and contributes to the increase of knowledge in the field of phytoestrogens and abdominal obesity.

We want to express our gratitude to the Reviewer for such positive reception of our work.

Reviewer 2 Report

It will be useful to mention at least briefly the function of some substances (and genes) in some chapters. People who are not complete experts in the field of human obesity would benefit from reading the article. For example : PPARs in chapter 2.1 and IL, leptin, IFNB1, inflamasoms a etc. in chapter 4.2 In the tables I was not quite well oriented, because I was not quite clear which data belong together (consists of 1 row). A minor note: in line 504 there is a typo, there is a reach instead rich.

Author Response

Reviewer 2

  • It will be useful to mention at least briefly the function of some substances (and genes) in some chapters. People who are not complete experts in the field of human obesity would benefit from reading the article. For example, PPARs in chapter 2.1 and IL, leptin, IFNB1, inflammasomes and, etc. in chapter 4.2

Following the Reviewer’s valuable suggestion, we have modified proper paragraphs within the text as follows:

"PPARs are nuclear receptors which are present in three isoforms: PPARα, PPARδ, and PPARγ, encoded by different genes. PPARs act as ligand-regulated transcription factors that control gene expression by binding to specific response elements within promoters. PPARs bind as heterodimers with a retinoid X receptor and, upon binding agonist, interact with cofactors that results in the increase of transcription rate [28]. Page 4, lines 144-149

"Genistein, via inhibition of nuclear factor κB (NF-κB, a protein complex that controls transcription of key genes involved, among others, in inflammatory response and apoptosis), downregulates synthesis of pro-inflammatory interleukins (e.g., IL-6 and IL-8) in mouse 3T3-L1 cells. At the same time, by inhibition of JAK-2 protein tyrosine kinase, genistein in human fibroblasts decreases expression of leptin – a hormone and a pro-inflammatory cytokine secreted by adipocytes [65]. Genistein-based supplementation also led to the activation of the upstream regulator interferon b1 (IFNB1). Activation of this cytokine inhibits the NLRP1b and NLRP3 inflammasomes (cytosolic multiprotein oligomers of the innate immune system responsible for the activation of inflammatory responses) via the upregulation of the signal transducer and activator of transcription factor 1 (STAT1) [94]. However, these effects were observed only in equol producers. Also, biochanin A, via inhibition of PPARγ and blocking MAPK phosphorylation, downregulates leptin, TNFα, and IL-6 expression in primary rat adipose-derived stem cells [71].” Page 9, lines 394-405

  • In the tables, I was not quite well oriented, because I was not quite clear which data belong together (consists of 1 row).

We apologize for this inconvenience, but during manuscript adaptation to the journal requirements, the tables were re-formatted. We have restored the original table format, and to facilitate the interpretation, we have introduced additional lines separating individual rows from each other.

  • A minor note: in line 504, there is a typo, there is a reach instead rich.

We apologize for this typo; the corrected sentence is as follows:

"In the Framingham Offspring Study, high dietary intake of isoflavones and lignans was associated with significantly lower plasma TG levels [103], while in a cross-sectional study assessing the influence of isoflavone intake on cardiovascular disease risk factors, diet rich in daidzein was positively associated with HDL cholesterol in postmenopausal women [104].” Page 15, lines 525-529.

Reviewer 3 Report

The work represents a complete and detailed overview of the topic. Reading is fluent and the literature data reported are adequate. I particularly appreciated paragraphs 4 and 5 which represent the core of the work itself.

However, it is necessary to pay attention to some steps to avoid that the manuscript can be penalized by incompleteness and gross errors. Following my suggestions:

Reference 6 is incorrect. This has probably generated a shift of all subsequent references, as I have been able to ascertain several times in the references along with the manuscript. I recommend to correct the error and carefully check all the references.

At the end of line 44, a reference would be useful. Lancet, 351 (9106), 853-6. Lean 1998 could be an option but something more recent would also be fine.

At line 60-64 the endocrine role of phytoestrogens is deepened. To be more in line with current literature evidence, and with the content of the manuscript later in the description, I think natural phytoestrogens are more correctly described as estrogen receptor modulators (STEAR or SERM), rather than chemicals endocrine disruptors. For reference, try the following, if they can be useful.

Crit. Rev. Biochem. Mol. Biol. 2002, 37, 1–28.

Curr. Med. Chem. 2003.10, 181-210.

Engl. J. Med. 2003, 348, 618–629.

Endocr. Rev. 2004, 25, 45–71.

The classification in paragraph 2 is very useful. I would suggest adding that the term "phytoestrogen" refers to a functional classification rather than a chemical one. The reference family remains that of polyphenols.

On line 88, I think the correct phrase is "concentrations of equol".

It may be useful to create a diagram summarizing the phytoestrogen classification as described in paragraph 1

On line 119 "with either" can be removed.

At line 129 it is better to replace "Are" with "seems to be", taking into account that reference is about unpublished data.

On line 131 delete "an"

At line 136 it might be useful to highlight the biphasic mechanism of phytoestrogens (Br J Nutr, 91 (4), 513-31. Magee 2004 as an example)

Lines 137 to 143 seem more appropriate for paragraph 2.2

At line 151 are we referring to humans?

At line 151, replace "can" with "seems".

I can't understand the concept expressed in lines 151-153 well. Can you develop it better?

At line 165 are we referring to studies in humans?

At line 170 are we referring to in vitro studies on human cellular models? In general, it is increasingly transparent to specify the origin of the data if it refers to animal models, in vitro with human or animal cells or directly in humans. I think it is an aspect that deserves attention for the paragraph 2 (other suggestions: lines 174, 198, 202, 222, 236, 295, 340, 378, 383), as was done adequately in the following paragraphs.

Probably, the model or organism can be mentioned in tables 1 and 2.

At line 223 it is better to express the concept as follows: "Polyphenols indirectly exert their ..."

At lines 223-238, the mechanism of modulation of miRNA by polyphenols could be better explained from a mechanistic point of view.

At line 497 change the term connects with links

Round 2

Reviewer 3 Report

The authors responded adequately to the suggestions and consistently modified the manuscript, improving it considerably.